# Enhancing Photoluminescence Intensity and Spectral Bandwidth of Hybrid Nanofiber/Thin-Film Multilayer Tm^3+^-Doped SiO_2_–HfO_2_

**DOI:** 10.3390/nano12213739

**Published:** 2022-10-25

**Authors:** Nurul Izzati Zafirah Zulfikri, Abdel-Baset M. A. Ibrahim, Nur Amalina Mustaffa, Rozan Mohamad Yunus, Suraya Ahmad Kamil

**Affiliations:** 1Faculty of Applied Sciences, Universiti Teknologi MARA, 40450 Shah Alam, Selangor, Malaysia; 2Fuel Cell Institute, Universiti Kebangsaan Malaysia, 43600 Bangi, Selangor, Malaysia

**Keywords:** nanofiber, thin film, photoluminescence, rare earth, optical materials, multilayering, thulium, silica-hafnia

## Abstract

Multilayering of optical thin films is widely used for a range of purposes in photonic technology, but the development of nanofiber structures that can outperform thin films and nanoparticles in optical applications cannot simply be disregarded. Hybrid structures composed of Tm^3+^-doped SiO_2_–HfO_2_ in the form of nanofibers (NFs) and thin films (TFs) are deposited on a single substrate using the electrospinning and dip-coating methods, respectively. Ultrafine nanofiber strands with a diameter of 10–60 nm were fabricated in both single and multilayer samples. Enhanced photoluminescence emission intensity of about 10 times was attained at wavelengths of around 457, 512 and 634 nm under an excitation of 350 nm for NF-TF-NF* hybrid structures when compared with single-layered NF and TF structures. The arrangement of nanofibers and thin films in a multilayer structure influenced the luminescence intensity and spectral bandwidth. High transparency in the range of 75–95% transparency across the wavelength of 200–2000 nm was achieved, making it ideal for photonic application. Theoretical findings obtained through IMD software were compared with experimental results, and they were found to be in good agreement.

## 1. Introduction

Thin-film deposition is a very common technique used in a variety of applications, including telecommunications, solar cells, integrated circuits, semiconductor devices, wireless communications, photoconductors and light crystal displays, light emitting diodes, transistors and other emerging technologies [1]. To meet the demands of modern technology, thin films have undergone numerous modifications throughout the years. One of the most well-known techniques that produces notable results is layering. In fact, multilayer optical thin films have been widely exploited to obtain specific optical characteristics for certain applications in a variety of fields of optical technology. Numerous studies on thin-film stacking using a variety of compositions, designs and approaches have been conducted. The general concept of multilayering thin-film structures is based on the fact that optical interference is made of two layers of high and low refractive indexes that are arranged alternately onto a single substrate [2,3,4]. Moreover, a previous study conducted by Rahmani and Ardyanian [5] discovered that as the number of thin-film layers of ZnO and TiO_2_ increases, the band gap of sample also increases, leading to a higher reflective index and absorbance. Thus, a multilayer structure is the best choice for photonic applications.

The optical applications of electrospun nanofibers have sparked much interest, because they can be used in sub-wavelength components for light generation, confinement, guiding and detection [6,7]. The excellent performance of nanofiber structures in terms of charge and energy transfer makes them preferable to nanoparticles and thin films due their unique physical properties; namely, high surface-to-volume ratio and high porosity. Owing to their physical properties, compact-sized optical circuits and photonic components can be constructed [8,9,10]. Moreover, due to their high-porosity structure, they can transport ions rapidly and are capable of aiding long-lasting electrolyte storage [11], resulting in a significant improvement in the efficiency of the energy-related nanofiber-based devices. Electrospun nanofibers have a refractive index of 1.5 to 2.5 at visible wavelengths, which is comparable with that of a normal optical fiber. When surrounded by air or a medium with a lower refractive index, they can direct light through total internal reflection [12].

Furthermore, nanofiber can be simply fabricated using electrospinning method as it is a straightforward technique that can control the nanofiber size and morphology to attain the preferred properties. The electrospinning method is the most favorable, because it is a cost-effective technique that can produce high surface-to-volume ratio, has tunable porosity and adaptability. The sol-gel method is among the common methods used in producing nanofiber solutions due to its versatile route for synthesizing inorganic and organic–inorganic structures, including glasses, ceramics and films [13].

Rare earth (RE) ions have attracted much attention in the development of optical amplifiers and solid-state lasers because of their sharp emission lines due to an electronic transition in the 4f band and their capacity to amplify weak signals [14,15,16]. Prior research has demonstrated that materials that include quantum dots and nanostructures can emit narrower photoluminescence (PL) peaks with a high quantum yield, larger absorption bands and significant effective Stokes shifts [17]. Qin et al. [18] demonstrated that the presence of an active RE ion in the 1D structure of Eu^3+^/Tb^3+^ co-doped LabBO_3_ nanofibers results in a luminescence intensity that is more intense and has a longer decay time compared with the powder sample of the same composites.

In the present work, we fabricated a multilayer nanofiber/thin-film structure composed of Tm^3+^-doped silica-hafnia. The idea of stacking both nanofiber and thin-film structures is to enhance the emission intensity without causing an unwanted effect, such as concentration quenching. Thus, the optical properties of the fabricated hybrid structures were analyzed, and a comparison with theoretical results, which was obtained through IMD software, is provided. Additionally, the morphological, structural and optical properties of thin films and nanofibers are also compared.

## 2. Materials and Methods

### 2.1. Solution Preparation

In this study, the amounts of SiO_2_ and HfO_2_ used were 90 mol% and 10 mol%, respectively. A sol-gel solution was firstly made by combining two solutions, which are referred to as solution A (sol A) and solution B (sol B). Sol A represents 20 mL of a silica solution that was made by mixing tetraethylorthosilicate (TEOS), ethanol (EtOH), ultra-pure water (H_2_O) and hydrochloric acid (HCl) with a molar ratio of TEOS:HCl:EtOH: H_2_O of 1:0.01:37.9:2. The role of HCl in sol A was to act as a catalyst, whereas EtOH acted as a solvent. The solution was then hydrolyzed for 1 h with a stirring rate of 400 rpm at 65 °C. At the same time, 20 mL of HfO_2_ solution (sol B) was prepared by mixing hafnium (IV) chloride (HfCl_4_) powder with ethanol (EtOH). The solution was stirred at the rate of 400 rpm until HfCl_4_ powder dissolved. Once sol A and sol B were done, aqueous thulium (III) chloride hexahydrate (TmCl_3_∙6H_2_O) was added into sol A. Sol B was then transferred into the sol A using a syringe. The resultant mixture was stirred for 16 h at room temperature with a stirring speed rate of 400 rpm. An additional step was required before depositing nanofibers, in which poly (vinyl) alcohol (PVA) solution was added into the sol-gel solution to produce continuous nanofiber strands.

### 2.2. Deposition of Nanofiber/Thin-Film Multilayer Structure

Nanofibers were deposited using a standalone NE-1000 Programmable Single Syringe Pump. The deposited samples were then annealed using a CARBOLITE Shimaden CWF 11/5 chamber furnace, in which (i) temperature was ramped up slowly at the rate of 5 °C/min to 950 °C, (ii) temperature was maintained at 950 °C for 1 h and (iii) temperature was then ramped down at the rate of 10 °C/min. Meanwhile, a single layer of the thin film comprising stacks of 20 nanoscale layers was deposited using a computerized KSV dip coater system. The dipping/withdrawal speed was kept constant at 40 mm/min. After the final nanoscale layer was deposited, the resulting film underwent heat treatment using CARBOLITE three-zone furnace for 30 min at 950 °C. The purpose of the annealing process is to eliminate the presence of the OH^−^ group, to improve the thin-film surface properties, and to strengthen the network structure through densification [19].

A few layers of nanofibers and thin films were deposited on a fused silica substrate and were arranged alternately. The deposited samples were identified as single-layered, nanofiber (NF), thin-film (TF), dual-layered NF–TF*, TF–NF* and three-layered NF–TF–NF* and TF–NF–TF*. The asterisk (*) symbol indicates the final top layer deposited on the substrate. Figure 1 illustrates the sequence of deposition of nanofibers and thin films on a fused silica glass substrate, as follows: (a) NF, (b) TF, (c) NF–TF*, (d) TF–NF*, (e) NF–TF–NF* and (f) TF–NF–NF*. Both structures were composed of 0.8 mol% of Tm^3+^-doped 90SiO_2_–10HfO_2_.

### 2.3. Sample Characterization

X-ray diffraction (PANalytical X’pert PRO, Almelo, The Netherlands) was used to examine and compare the crystallinity of the single-layered nanofibers and thin films, and Fourier transform infrared spectroscopy (FTIR) (PerkinElmer Spectrum One, Wellesley, MA, USA) was employed to obtain the infrared absorption spectra of the produced materials. Field-emission scanning electron microscopy (FESEM, Joel JSM-7600F, Tokyo, Japan) was utilized to examine the morphology of the electrospun nanofiber samples, and Image J software (Version 1.53k, U.S. National Institutes of Health, Bethesda, MD, USA) was used to determine the diameter of the deposited nanofibers. The elemental composition of the samples was determined using energy-dispersive X-ray spectroscopy (EDX). The optical transparency of the samples was evaluated using a UV–vis NIR spectrophotometer (Varian Cary 5000, Palo Alto, CA, US) at a range of 200–2000 nm. By using a xenon lamp as an excitation source with a wavelength of 350 nm, PL spectroscopy (FLS920 Edinburgh Instrument, Livingston, UK) was employed to determine the luminescence spectra.

### 2.4. Theoretical Aspects

The optical characteristics of the deposited samples were compared with theoretical results. The experimental results of transmission and absorbance of the samples were compared with the theoretical results obtained using IMD software, (Version 5.04, Bell Laboratories, Murray Hill, NJ, USA) whereas the PL emission intensity data were fitted to the Voigt function. The IMD software calculated the reflection, transmission and absorption of an electromagnetic plane wave at the interface of a multilayer structure based on the transfer matrix method (TMM) [20]. Fresnel’s equations for *s*-polarization, where electric-field amplitude, *E*, is perpendicular to the plane of incidence, were as follows:(1)$$r_{s}=\frac{\left|E_{r}\right|}{\left|E_{i}\right|}=\frac{n_{i}\cos\theta_{i}-n_{t}\cos\theta_{t}}{n_{i}\cos\theta_{i}+n_{t}\cos\theta_{t}},$$
(2)$$t_{s}=\frac{\left|E_{t}\right|}{\left|E_{i}\right|}=\frac{2n_{i}\cos\theta_{i}}{n_{i}\cos\theta_{i}+n_{t}\cos\theta_{t}},$$

Fresnel equations of *p*-polarization, where *E* is parallel to the plain of incidence, were as follows:(3)$$r_{p}=\frac{\left|E_{r}\right|}{\left|E_{i}\right|}=\frac{n_{i}\cos\theta_{t}-n_{t}\cos\theta_{i}}{n_{i}\cos\theta_{t}+n_{t}\cos\theta_{i}}$$
(4)$$t_{p}=\frac{\left|E_{t}\right|}{\left|E_{i}\right|}=\frac{2n_{i}\cos\theta_{i}}{n_{i}\cos\theta_{t}+n_{t}\cos\theta_{i}}$$
where *r* represent the reflection coefficient and *t* is the coefficient of transmission. The refractive indices of a medium of incident light and transmitted light are denoted as *n_i_* and *n_t,_* respectively. The optical function of the multilayer stack was obtained by the derivation of Fresnel’s equation by considering the thickness, interfacial roughness/diffuseness and optical constant of each layer.

## 3. Results and Discussion

### 3.1. Comparing Thin-Film and Nanofiber Results

The X-ray interference patterns of single-layered nanofiber and thin-film structures were compared, as shown in Figure 2a. The patterns of the two structures were identical, and neither showed a prominent high peak of crystallization, suggesting that the produced samples were in the amorphous phase. Both NF and TF exhibited the same two noticeable humps, which were situated at 2θ = 10° and 22°. However, NF samples had a substantially higher intensity than TF samples, which could be due to the present of PVA in the nanofiber samples. The appearance of such diffraction patterns in PVA, according to Gupta et al. [21], can be attributed to the presence of a very minor degree of crystallinity, which suggested the coexistence of a small PVA nanocrystalline phase with the amorphous phase material.

The OH^−^ group was not present in the single-layered TF and NF structures based on the FTIR analysis of the Tm^3+^-doped SiO_2_–HfO_2_ samples shown in Figure 2b, as no apparent peaks were seen in the range of 3700–3200 cm^−1^. Peaks in the TF sample were identical to those in the NF sample, suggesting that the bonds that existed in both samples were the same. Both samples had peaks located at 740, 614 and 450 cm^−1^ assigned to the presence of Hf–O bond, which was around 700–400 cm^−1^ [22,23]. Based on Figure 2b, both TF and NF samples had prominent peaks at around 810 and 1200 cm^−1^, corresponding to the longitudinal optical (LO), whereas peaks located at 1100 and 1180 cm^−1^ were assigned to transverse optical (TO) components of asymmetric stretch of the SiO_4_ unit [22,23,24]. Furthermore, both TF and NF samples also had a shoulder at around ~970 cm^−1^, which corresponded to Si–O–Hf bond [23,25]. An additional peak was observed at ~2260 cm^−1^, which was assigned to Si–H stretching [26], and this may be related to TEOS, as silicon alkoxide precursor may result in SiH after hydrolysis and condensation of the sol-gel process [27,28]. During the sol-gel process, the partial oxidation of Si–H bonds might result in the formation of SiO_4_ units through the formation of new Si–O bonds [29,30].

The emission spectra of a single-layered Tm^3+^-doped SiO_2_–HfO_2_ thin-film and nanofiber structure under excitation of 350 nm are shown in Figure 3. The full width at half maximum (FWHM) was labeled with a blue arrow. Both nanofiber and thin film structures were able to emit distinctive emission around 457, 512 and 634 nm, which corresponded to ^1^D_2_ → ^3^F_4_, ^1^D_2_ → ^3^H_5_ and ^1^G_4_ → ^3^F_4_ transition, respectively. A few weak emission peaks were observed at around 578 nm, corresponding to ^1^D_2_ → ^3^H_5_ transition. Despite having the same amount of Tm^3+^ ion, the emission produced by nanofibers was at a substantially higher intensity than that of the thin-film structure.

Based on Figure 3, the emission peaks produced by the thin-film structure demonstrated broader spectral bandwidth compared with nanofiber. As previously stated, the high-emission intensity of the NF sample was mostly attributable to its physical features, i.e., its high surface-to-volume ratio. The excited spontaneous emission in the light-emitting nanofibers can partly be explained as waveguides moving along the fibers and being transmitted into neighboring fibers, thus causing the emission produced to increase. This finding was also proven by previous studies, in which the absorption rate of the incident photons became greater when compared with thin films, because the excitation light was propagated through nanofibers [14,31]. Moreover, Sun [32] stated that when particle size decreased, the band gap enlarged, resulting in an increment in the photoluminescence (PL) and photoabsorption (PA) spectra.

### 3.2. Morphological and Structural Properties of Multilayer Structures

A smooth and crack-free thin film of Tm^3+^-doped SiO_2_–HfO_2_ was produced in all multilayer structures even after being annealed at 950 °C, as shown in Figure 4. All multilayer structure samples of the deposited electrospun nanofibers had uniform, smooth fibers with random orientation and ultrafine strands with a range diameter of 10–60 nm. Figure 5 shows the cross-section of the TF-NF-TF* multilayer structure. The thickness of the deposited thin film was around 1.85 μm, whereas the thickness of the nanofiber layer was approximately 100–120 nm. The thin film was expected to be much thicker compared with the nanofiber layer, as it contained a 20-nano layer of thin film.

Figure 6 shows the elemental analysis of the NF-TF-NF* multilayer structure comprised of Tm^3+^-doped SiO_2_–HfO_2_, which was performed using EDS. The desired elements, Si, Hf, O and Tm, were present in the sample, whereas the appearance of element C in the EDS result was due to the carbon tape needed for FESEM-EDS analysis. Table 1 summarizes the elemental composition of samples of NF and NF-TF-NF*. The NF-TF-NF* sample contained more Tm than the NF sample, showing that the Tm^3+^ ion was effectively doped into each deposited layer of the NF-TF-NF* sample. Figure 7 depicts the EDS mapping of the individual element present in the scanned area, which confirmed that the ions were equally distributed throughout the host matrix.

### 3.3. Optical Properties of Multilayer Structures

The obtained photoluminescence emission peak of single-layered and multilayer structure samples is depicted in Figure 8. All samples were able to produce similar major peaks, which were around 457, 512 and 634 nm. The photoluminescence intensity of nanofibers significantly increased when paired with thin-film structures. From the obtained results, NF-TF-NF* had the most significant PL emission, followed by TF-NF-TF*, TF-NF*, NF-TF*, NF and finally TF. This indicated that the emission peak increased with an increasing number of layers of nanofiber/thin film. This also suggested that the increase in emission intensities was caused by the emergence of nanoscale nanofiber structures, which amplified emission intensities, whereas the increase in spectral bandwidth was caused by the presence of 2D thin-film structures. In earlier studies, Abd-Rahman and Razaki [14] discovered that the embodiment of nanostructured layers, which changed the energy levels of Tm^3+^ due to the confinement of the ions in a low-dimensional structure, combined with the effect from thin film, leading to the spectral expansion and enhancement of emission intensity.

The results obtained showed that the emission intensity of TF-NF* was higher compared with that of NF-TF*. This occurrence was probably due to the light trapped in the active nanofiber layer on the thin-film layer. Prior research by Chang et al. [7] revealed that the major enhancement of the emission intensity was related to the prolonged light path of the absorbed incoming light in the nanofiber layer [14], which caused multiple scattering events to occur inside the nanofiber layer. However, TF-NF-TF* still had a lower emission intensity when compared with NF-TF-NF* despite having nanofiber trapped between active thin films, indicating that the involvement of a nanofiber layer played a huge role in producing significantly high-emission intensity.

Furthermore, NF-TF-NF* multilayer structure was discovered to have the largest intensity peak, which was about 10 times more than that of the single-layered structure. This was believed to be the outcome of an increase in active ion interactions, which increased the chance of net photon capture during the excitation process [14,31]. Moreover, this response was mediated by the surface contacts between fibers, thereby indicating that the two nanofiber layers in the sample were the main contributors to the enhancement of the emission intensity of the NF-TF-NF* sample. A few weak peaks were observed at around the 550–600 region (Figure 8b), which might correspond to ^1^D_2_ → ^3^H_4,5_ and ^1^G_4_ → ^3^F_4_ transition [33]. Figure 9 illustrates the mechanism of light excitation that propagated through active fiber strands in the nanofiber layers, thereby showing that the absorption points in nanofibers were much higher compared with that in the thin-film structures due to the high surface-to-volume of the nanofiber structures. This finding proved that the higher the absorption was, the higher the number of molecules promoted to the excited state was, thereby causing higher emission production.

The transmittance of single-layered and multilayer structures at a region of 200–2000 nm is demonstrated in Figure 10. All fabricated samples exhibited a drastic increment of optical transparency in the UV–visible wavelength, whereas in the infrared region (700–2000 nm) the transparency remained relatively constant. The sample that had the highest transmittance percentage was NF, followed by TF, NF-TF*, TF-NF*, TF-NF-TF* and NF-TF-NF*. This demonstrated that the percentage of transmittance decreased as the number of layers increased, which was expected since the resulting sample structure becomes thicker. However, NF-TF-NF* was discovered to have a lower transmittance than TF-NF-TF*, despite its smaller resultant thickness. This was probably due to light scattering that contributed to the decline of transmittance percentage of the NF-TF-NF* structure. The low transmittance of the NF-TF-NF* sample was mainly due to the large surface-to-volume ratio of the nanofiber structure, which caused light to deflect in all directions, thus leading to the increment of the absorption rate on a single substrate. Considering that the NF-TF-NF* was composed of two nanofiber layers that “sandwiched” the thin-film layer, a higher absorption rate was obtained. Nevertheless, single-layered NF was still the most transparent, because it had a lower density and was far thinner than a single-layered thin film.

### 3.4. Comparison of Experimental and Theoretical Results

The experimental data were compared with the multilayer stack computation produced by the IMD software. The thickness of thin films was fixed at 1.80 μm, whereas the thickness of nanofibers varied from 100–120 nm. Other parameters involved, such as optical constant, atomic weight, composition and density of each material, were taken from data stored in the IMD software and were kept constant. The transmittance results, both experimental and theoretical, of (a) single-layered, (b) dual-layered and (c) three-layered structures in the range of 200–2000 nm are shown in Figure 11. The pattern of the calculated transmittance shows a similar result to the experimental data of Tm^3+^-doped SiO_2_-HfO_2_. The only difference was that the slope of the calculated transmittance gradually increased in the UV region towards the IR region, whereas the experimental result showed that the transmittance increased drastically towards the IR region before remaining constant throughout the IR region. This finding signified that the fabricated multilayer structure obeyed the modified Fresnel’s equation where the interface imperfections were considered.

Similar to the experimental result, the transmittance decreased with the number of active layers due to light scattering. Killada [34] mentioned that the amount of light that is transmitted is influenced by the material’s thickness and the way that photons interact with the structure of the material. Additionally, HfO_2_ leads to the high refractive index of the layers. An increase in the refractive index concurrent with an increase in thickness demonstrates the dense uniformity and adhesion of both deposited structure layers. As a result, the absorbance and refractive index of the sample increases along with the number of layers, causing the percentage of light transmitted through the sample to be lowered, except for in the NF-TF-NF* structure, as discussed earlier.

The absorbance was also estimated using IMD software, and the results are as shown in Figure 12. In this study, the roughness of nanofibers was varied until they achieved the optimum degree, which was around 14.0–15.0 nm. Surface roughness significantly influences the rate of light absorption, as high roughness causes less light to reflect and more light to penetrate into the active layers. According to Liaparinos [35], the degree of roughness influences surface behavior, which affects the overall optical performance of the system. Based on Figure 12, the results obtained were similar to the experimental results. For single-layered samples, the absorbance of TF was higher compared with that of NF; however, it was notable that as the number of layers increased, the absorbance of the sample that consisted of nanofibers on the topmost layer had higher absorbance.

Therefore, a great variation of performance was present, and this depended on the resultant thickness of the sample and the surface roughness. The theoretical prediction and the experimental findings had a reasonable degree of coincidence. Photoluminescence intensity was fitted using the Voigt function. Figure 13 shows PL emission intensity data (dotted line) and the Voigt function profile (solid colored line). The parameters of the calculated data are tabulated in Table 2.

Based on Figure 13, the Voigt function provided the best match for the PL emission peak produced from the experiment, with just a small difference. Hence, the fitted graphs were acceptable. Based on tabulated data in Table 2, the peak that was present at 600–680 nm in the NF samples had a bigger area under the graph (peak 5 = 10,401 a.u.) when compared with the emissions of TF samples (peak 5 = 4854 a.u.). This finding showed that the low dimensional size indeed increased the emission intensity. However, the emission peaks emitted by the TF samples were broader compared with that of the NF samples. For instance, peak 5 of the NF sample had a FWHM of 29.9 nm, whereas that of the TF sample was 31.3 nm, which was 1.4 nm higher. The same held for peak 1 of the NF sample with a FWHM of 21.4 nm, which was 4.6 nm narrower compared with peak 1 of the TF sample (26.0 nm).

The emission peak of the TF-NF* sample had a larger emission intensity when compared with that of the NF-TF* sample. Based on the computed data, the total area under the graph of the TF-NF* sample (peak 1 + peak 2 + peak 3 = 489,383 a.u.) at 430–550 nm region was 318,224 a.u. larger compared with that of the NF-TF* sample (peak 1 + peak 2 + peak 3 = 171,159 a.u.). In fact, the emission emitted at 600–680 nm in the TF-NF* sample (peak 5 = 101,948 a.u.) also had a bigger area under the graph, with a difference of 63,779 a.u. when compared with that of the NF-TF* sample (peak 4 = 38,169 a.u.). The FWHM of peak 5 of the TF-NF* sample (35.9 nm) was almost three times bigger when compared with the FWHM of peak 4 of NF-TF* (12.9 nm) samples. This finding showed that having nanofibers on the topmost layer prolonged the light path of the absorbed incoming light in the nanofiber layer compared with a single layer of nanofibers (NFs). However, when nanofibers were added onto NF-TF*, which produced NF-TF-NF*, the emission intensity was two times bigger compared with that of TF-NF-TF*.

The NF-TF-NF* multilayer structure had a larger emission intensity compared with other samples, as the areas under the graph for all peaks were larger compared with other peaks present in other samples. This finding was proven when compared with TF-NF-TF*, where both emissions had broad emission peaks at around 430–550 nm. NF-TF-NF* had three peaks (peaks 1, 2 and 3) at the mentioned emission region. The total areas under the graph of each peak were added, thereby producing an area of 704,968 a.u. Similar to NF-TF-NF*, TF-NF-TF* also had three peaks (peaks 1, 2 and 3) and had a total area under the graph of 407,188 a.u. When single-layered samples were compared with three-layered samples, the area under the graph of the single-layered samples increased by a factor of about 10, indicating that a significant increase in emission intensity was possible as the amount of multilayer increased. Hence, it was theoretically shown that the intensity of the PL emission generated increased with the number of layers in the multilayer structure. However, the FWHM of TF-NF-TF* was the widest among the samples. This was observed at the 600–680 nm region, where the FWHM of peak 4 of the TF-NF-TF* sample was 46.5 nm, which was 12.1 nm bigger when compared with the FWHM of peak 4 of the NF-TF-NF* sample (34.4 nm). This occurrence was likely due to the involvement of two thin-film layers, which caused the emissions to have a wide FWHM. The broad emission intensity could also be due to the presence of an overlapping emission peak at that particular region.

## 4. Conclusions

Significantly enhance PL emission intensity was successfully achieved owing to the structural properties of both nanofiber and thin-film structures and also a high amount of active ions doped in a single substrate. A comparison between nanofibers and thin films shows that nanofibers are able to produce higher emission intensity compared with thin films due to the following: (i) nanofibers had higher absorption points due to their high surface-to-volume ratio and (ii) the prolonged light path of the absorbed incoming light in the nanofiber layer. Nanofibers produced a narrow emission intensity compared with thin films. Therefore, specific applications were possible when constructing a multilayer structure and choosing the composite material. Moreover, the experimental and computed results reached a satisfactory agreement with accurate values of parameters such as the emission peak, FWHM and area under the graph.

## Figures and Tables

**Figure 1 nanomaterials-12-03739-f001:**
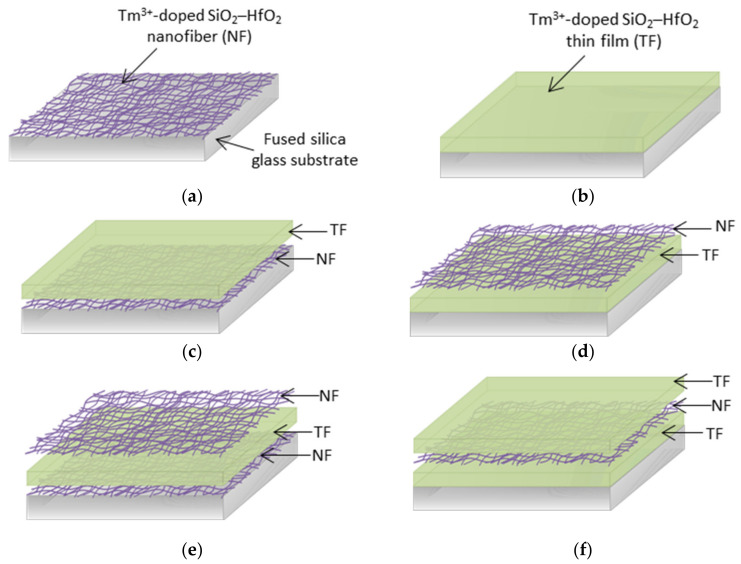
The sequence of deposition of nanofibers and thin films on fused SiO_2_ glass substrate: (**a**) NF, (**b**) TF, (**c**) NF–TF*, (**d**) TF–NF*, (**e**) NF–TF–NF* and (**f**) TF–NF–NF*.

**Figure 2 nanomaterials-12-03739-f002:**
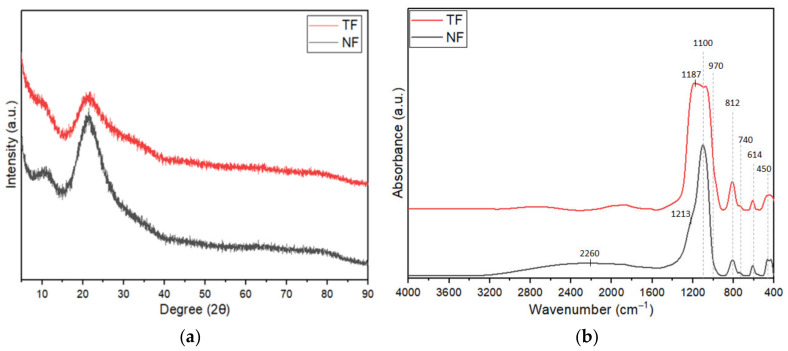
Both NFs and TFs were analyzed using (**a**) XRD and (**b**) FTIR.

**Figure 3 nanomaterials-12-03739-f003:**
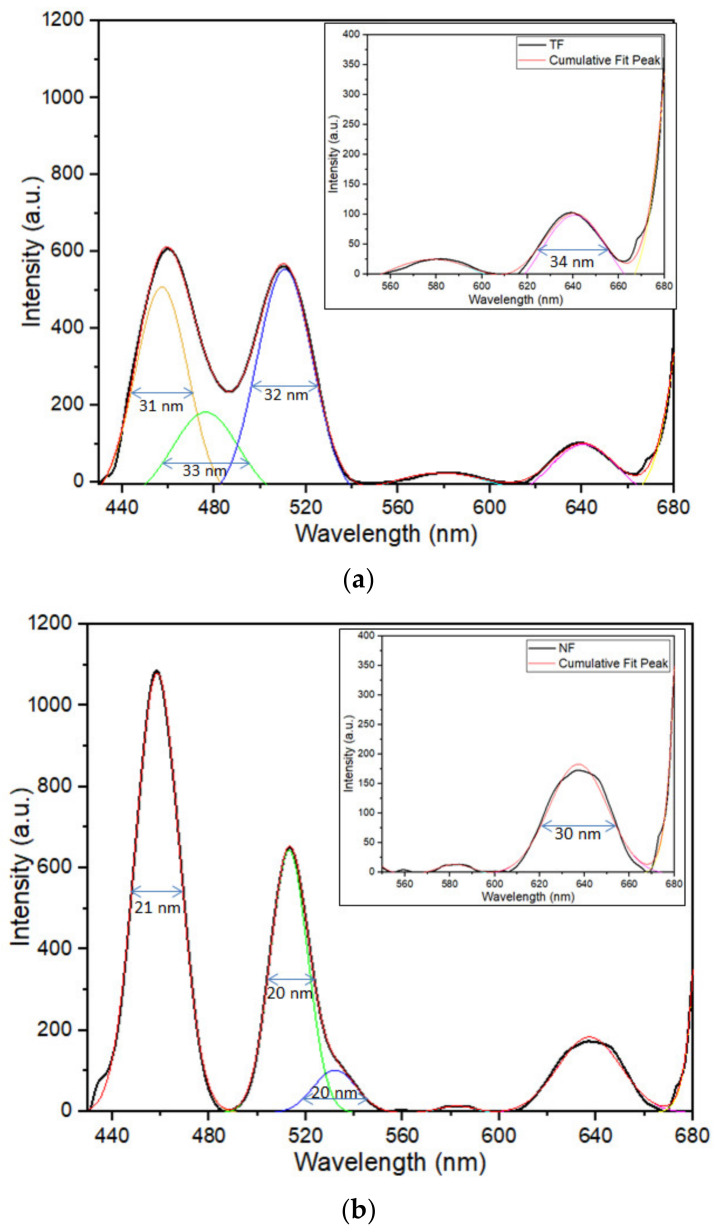
The emission spectra and full width at half maximum (FWHM) of Tm^3+^-doped SiO_2_–HfO_2_ single-layered (**a**) thin-film (TF) and (**b**) nanofiber (NF) structures under excitation of 350 nm. Inset shows the emission around 578 and 634 nm. Coloured lines (yellow, green, blue and pink) represents the convoluted peak that were present in emission spectra (black line), whereas red line is the cumulative fit peak.

**Figure 4 nanomaterials-12-03739-f004:**
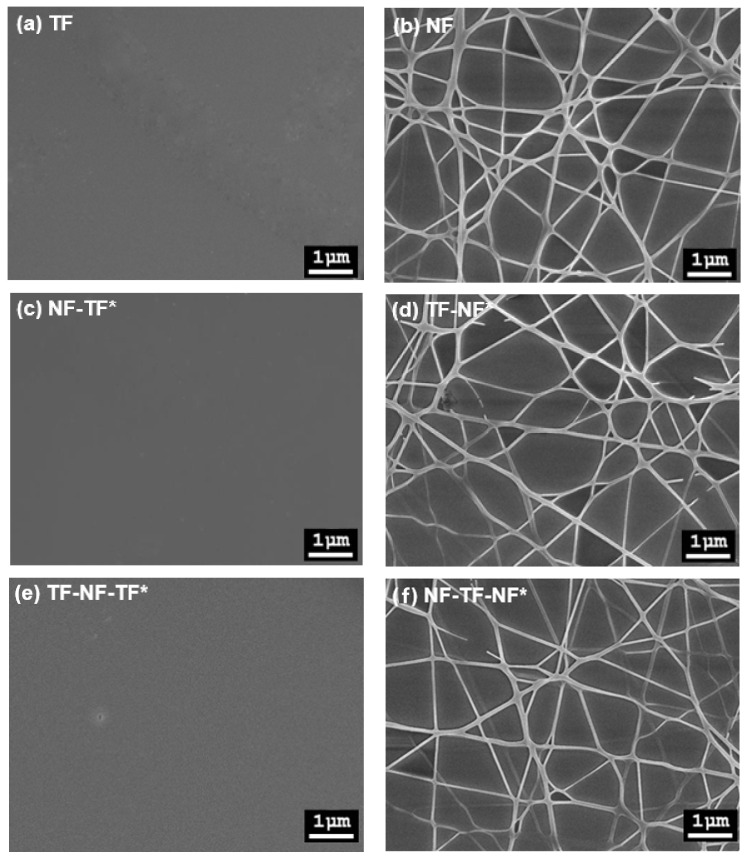
The field emission field scanning electron microscopy (FESEM) images of the top view of a multilayer structure sample at magnification of 15 kX: (**a**) TF, (**b**) NF, (**c**) NF-TF*, (**d**) TF-NF*, (**e**) TF-NF-TF* and (**f**) NF-TF-NF*.

**Figure 5 nanomaterials-12-03739-f005:**
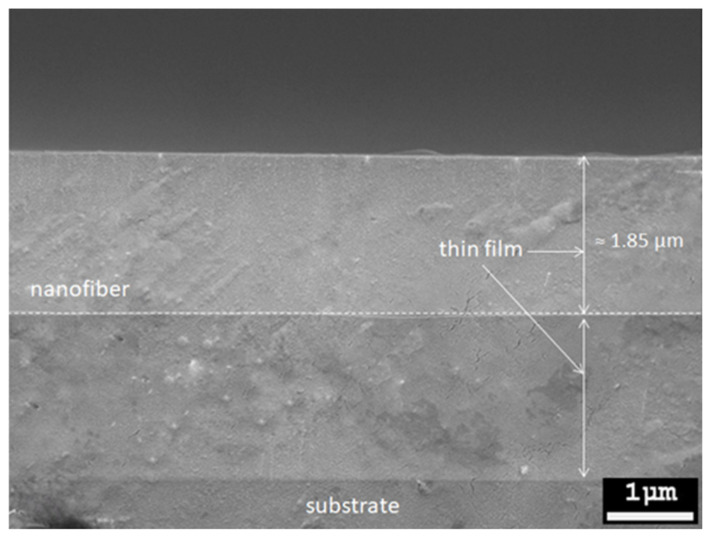
Cross-section of the TF-NF-TF* multilayer structure.

**Figure 6 nanomaterials-12-03739-f006:**
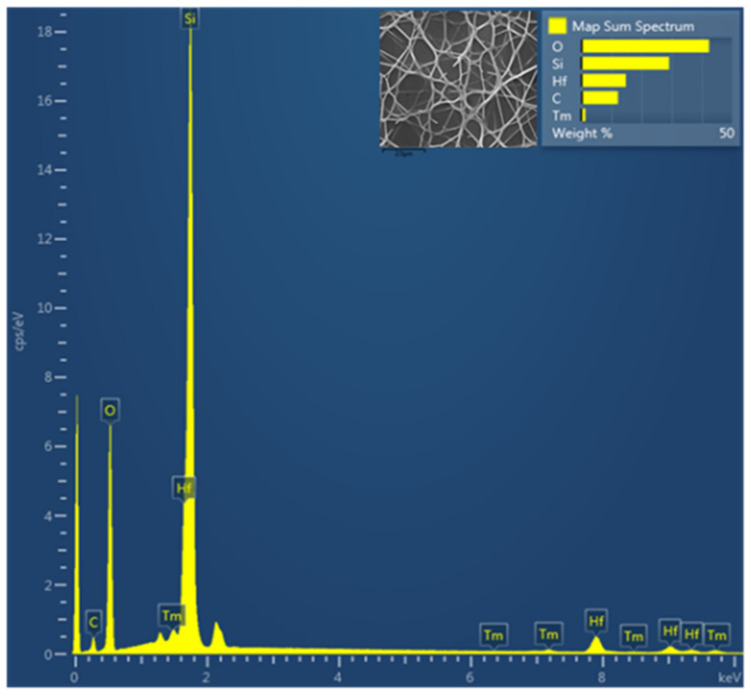
The energy dispersed spectroscopy (EDS) result of NF-TF-NF* multilayer structure Tm^3+^-doped SiO_2_-HfO_2_. The inset image is the area scanned.

**Figure 7 nanomaterials-12-03739-f007:**
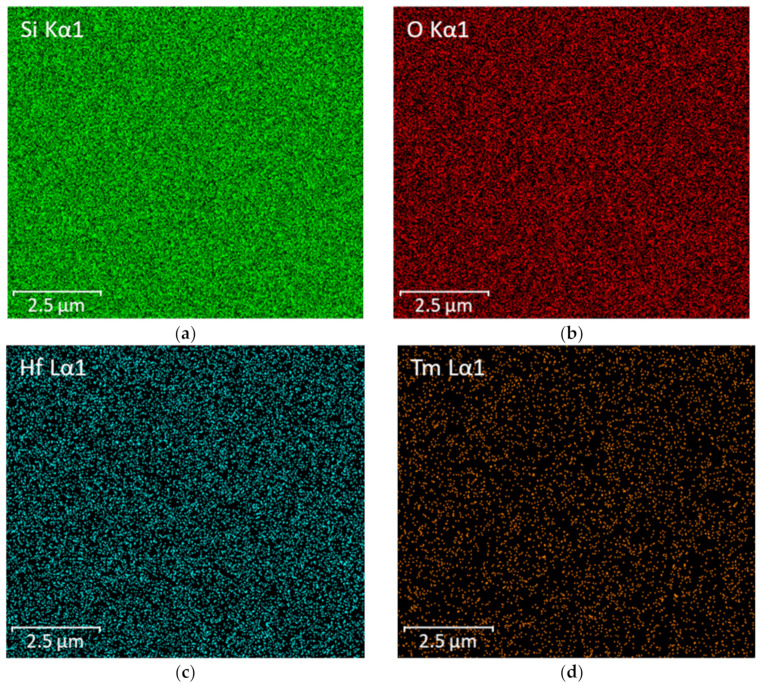
The element distribution of (**a**) Si, (**b**) O, (**c**) Hf and (**d**) Tm ions in the NF-TF-NF* sample.

**Figure 8 nanomaterials-12-03739-f008:**
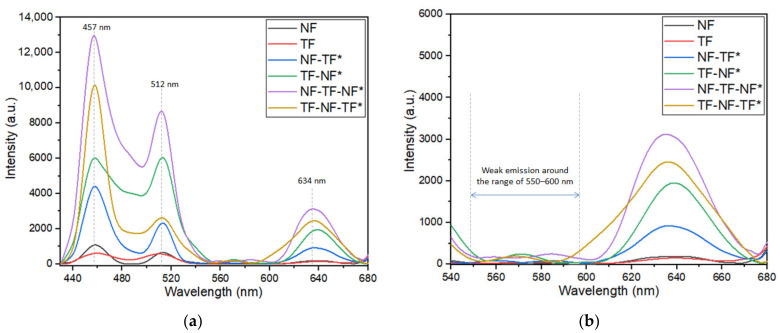
Emission spectra of Tm^3+^-doped SiO_2_–HfO_2_ multilayer structure at range of (**a**) 430–680 nm and (**b**) 540–680 nm under an excitation of 350 nm.

**Figure 9 nanomaterials-12-03739-f009:**
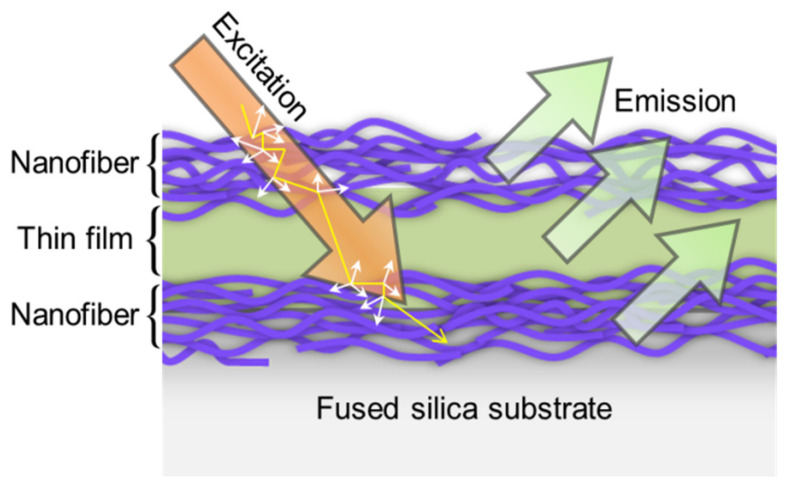
Mechanism of light excitation propagates through active fiber strands in nanofiber layers.

**Figure 10 nanomaterials-12-03739-f010:**
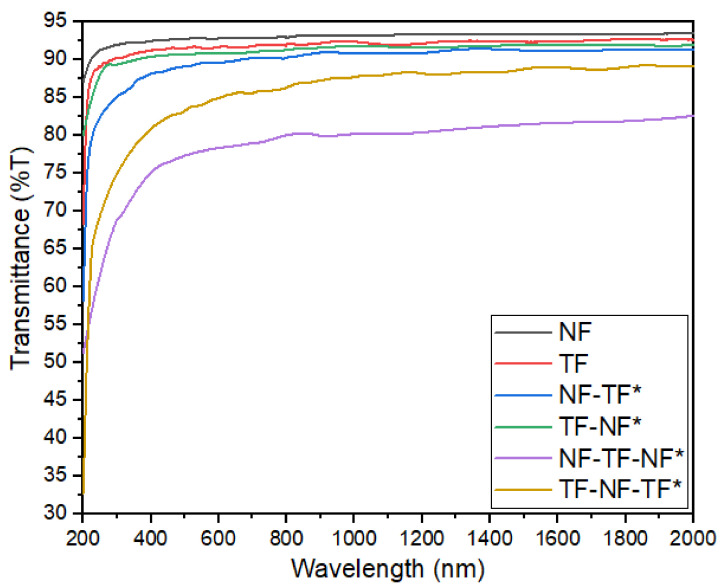
Transmittance of single-layered and multilayer structures at 200–2000 nm region.

**Figure 11 nanomaterials-12-03739-f011:**
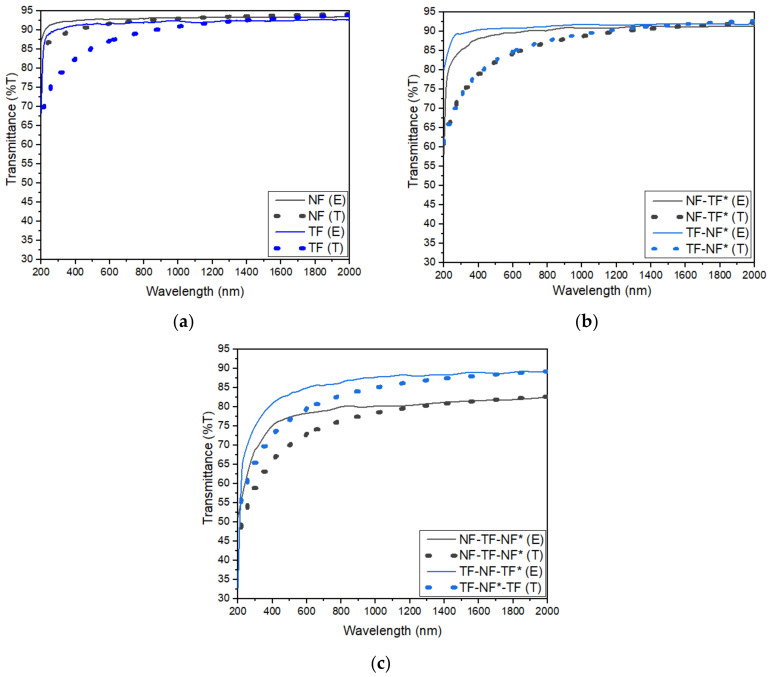
Transmittance of both experimental (E) and theoretical (T) of (**a**) single-layered structure, (**b**) dual-layered structure and (**c**) three-layered structure at the range of 200–2000 nm.

**Figure 12 nanomaterials-12-03739-f012:**
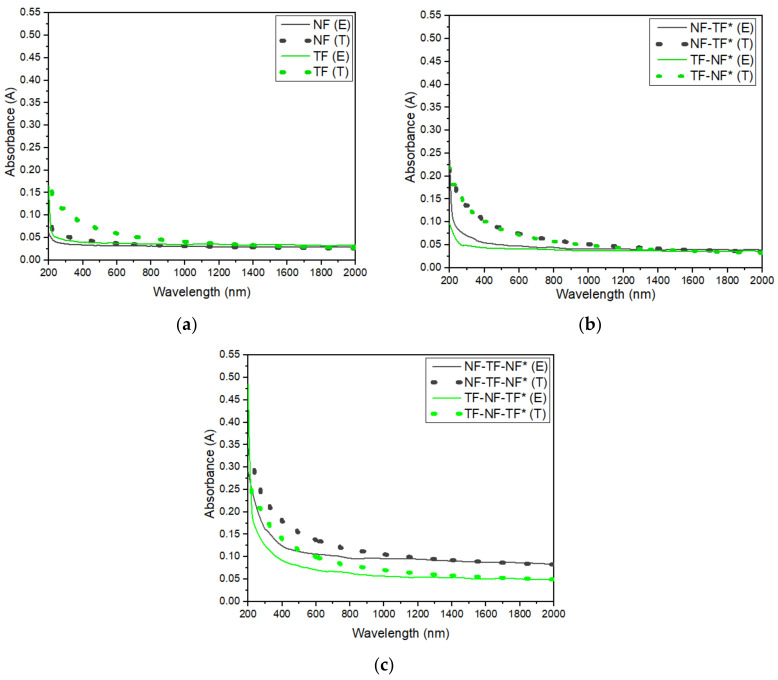
Absorbance of both experimental (E) and theoretical (T) of (**a**) single-layered structure, (**b**) dual-layered structure and (**c**) three-layered structure at the range of 200–2000 nm.

**Figure 13 nanomaterials-12-03739-f013:**
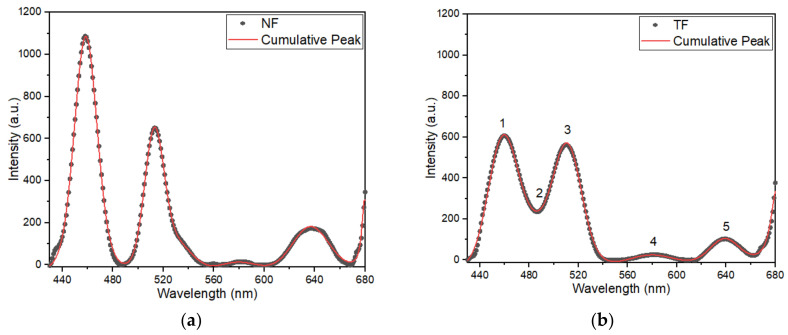
PL emission intensity data (dotted line) and the Voigt function profile (solid colored line) of (**a**) NF, (**b**) TF, (**c**) NF-TF*, (**d**) TF-NF*, (**e**) NF-TF-NF* and (**f**) TF-NF-TF* samples.

**Table 1 nanomaterials-12-03739-t001:** The elemental composition of NF and NF-TF-NF* samples.

Element (Line Type)	Atomic%	
	NF	NF-TF-NF*
C K	8.10	9.04
O K	47.13	40.18
Si K	27.71	28.63
Tm L	0.99	1.67
Hf M	16.07	20.48

**Table 2 nanomaterials-12-03739-t002:** Parameters obtained from Voigt function profile.

Sample	No. of Peak	Peak Centered (nm)	Area (a.u.)	Gaussian FWHM, wG (nm)	Lorentzian FWHM, wL (nm)	FWHM (nm)
NF	1	458.51 ± 0.07	27,398 ± 1698	19.13 ± 1.16	4.12 ± 2.13	21.4
	2	513.08 ± 0.43	13,052 ± 1140	18.56 ± 0.75	0	18.6
	3	532.95 ± 2.64	2528 ± 2835	20.68 ± 18.82	0	20.7
	4	580.41 ± 1.87	1132 ± 377	30.13 ± 5.49	0	30.1
	5	637.36 ± 0.46	10,401 ± 2629	9.74 ± 13.38	26.54 ± 10.68	29.9
TF	1	458.74 ± 1.66	17,969 ± 7419	25.99 ± 2.14	0	26.0
	2	485.11 ± 3.07	5888 ± 2341	30.91 ± 41.94	2.77 ± 1.68	32.4
	3	511.22 ± 1.46	18,428 ± 3535	25.02 ± 2.97	3.25 ± 3.29	26.8
	4	580.05 ± 2.05	3139 ± 1140	48.32 ± 5.21	0	48.3
	5	637.53 ± 0.46	4854 ± 4224	27.25 ± 14.24	7.23 ± 33.02	31.3
NF-TF*	1	458.87 ± 0.20	104,418 ± 3352	21.91 ± 0.97	0.68 ± 1.39	22.3
	2	511.82 ± 0.12	53,223 ± 2962	16.31 ± 1.42	5.67 ± 2.15	19.5
	3	480.16 ± 1.13	13,518 ± 2126	18.26 ± 1.98	25.69 ± 5.07	35.6
	4	637.91 ± 0.36	38,169 ± 4925	0	12.85 ± 7.54	12.9
TF-NF*	1	455.65 ± 0.10	91,357 ± 7848	21.69 ± 0.57	0	21.7
	2	477.17 ± 1.06	176,679 ± 11,901	43.43 ± 2.04	0	43.4
	3	513.86 ± 0.08	221,347 ± 5633	6.44 ± 1.72	23.24 ± 1.07	25.0
	4	575.29 ± 0.81	4302 ± 2304	17.17 ± 8.34	0.13 ± 15.30	17.2
	5	640.07 ± 0.13	101,948 ± 8665	25.73 ± 2.47	16.86 ± 4.15	35.9
NF-TF-NF*	1	455.88 ± 0.15	234,669 ± 15,351	20.95 ± 0.38	0	20.9
	2	478.79 ± 0.92	241,217 ± 19,747	36.35 ± 2.52	0	36.3
	3	512.73 ± 0.15	229,082 ± 6521	17.04 ± 0.78	9.52 ± 1.02	22.7
	4	637.27 ± 0.15	118,194 ± 6626	33.35 ± 1.92	2.02 ± 3.44	34.4
TF-NF-TF*	1	457.34 ± --	209,130 ± --	17.843 ± --	2.29 ± --	19.1
	2	475.96 ± 3.50	41,046 ± --	28.86 ± 20.02	0	28.9
	3	510.91 ± 1.36	157,012 ± --	5.0485 ± --	34.67 ± --	35.4
	4	637.43 ± 2.59	164,533 ± 35,765	31.57 ± 63.36	24.25 ± 122.79	46.5

## Data Availability

Not applicable.

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
