# Peer review of "Enhancing Photoluminescence Intensity and Spectral Bandwidth of Hybrid Nanofiber/Thin-Film Multilayer Tm3+-Doped SiO2–HfO2"

_nanomaterials, 2022, doi:10.3390/nano12213739_

Round 1
Reviewer 1 Report
In this work, the authors studied the PL properties of multilayer Tm3+-doped SiO2-HfO2. They observed that the NF-TF-NF multilayer hybrid structure showed ten times enhanced PL around the wavelengths of 457nm, 512nm, and 634nm. Furthermore, the authors observed that the NF arrangement affected the PL intensity and spectral bandwidth. This work is insightful and relevant to the readers of Nanomaterials, and can be published after some minor modifications below:
(1) In Fig. 3, fitting of the convoluted emission peaks at ~457nm and ~512nm should be shown for the TF sample. This is important to accurately obtain the FWHM.
(2) The Voigt function seems to be a poor fit to the PL spectrum in Fig. 13 between the 540-610nm region. See especially Fig. 13a and 13b. Can the authors explain why the fit is poor within this wavelength region?
(3) The y-axis of Fig. 11a-c and Fig. 12a-c should be plotted using the same scale for easy cross-comparison.
(4) In Table 2, the fitted parameters can be given in 3 or 5 significant figures for better readability.
Author Response
Response to Reviewer 1 Comments
In this work, the authors studied the PL properties of multilayer Tm3+-doped SiO2-HfO2. They observed that the NF-TF-NF multilayer hybrid structure showed ten times enhanced PL around the wavelengths of 457nm, 512 nm, and 634 nm. Furthermore, the authors observed that the NF arrangement affected the PL intensity and spectral bandwidth. This work is insightful and relevant to the readers of Nanomaterials, and can be published after some minor modifications below:
Point 1: In Fig. 3, fitting of the convoluted emission peaks at ~457nm and ~512nm should be shown for the TF sample. This is important to accurately obtain the FWHM.
Response 1: Correction has been made. The fitting of the convoluted emission of TF sample (Fig. 3(a)) has been added and compared with NF sample (Fig. 3(b)) in order to avoid confusion for the readers (Page 6).
Point 2: The Voigt function seems to be a poor fit to the PL spectrum in Fig. 13 between the 540-610nm region. See especially Fig. 13a and 13b. Can the authors explain why the fit is poor within this wavelength region
Response 2: The Voigt function of Fig 13(a) and (b) were fitted again and a better result was able to achieve (Page 14).
Point 3: The y-axis of Fig. 11a-c and Fig. 12a-c should be plotted using the same scale for easy cross-comparison
Response 3: Scale for graph in Fig 11(a)–(c) and Fig. 12 (a)–(c) has been fixed for easy cross-comparison (Page 13 and 14).
Point 4: In Table 2, the fitted parameters can be given in 3 or 5 significant figures for better readability.
Response 4: Changes have been made for Table 2 as advised by the reviewer (page 15-16).
Please see the attachment.

Reviewer 2 Report
The authors prepared different samples containing single layers and multi-layer structures of thin films and electrospun nano-fibers of a mixed SiO2 and HfO2 glass doped with Tm3+ ions. Due to the doping, the samples showed several photoluminescence bands in the visible part of the spectrum. The most important outcome is that in the multi-layer samples containing both thin films and nano-fiber layers, the electroluminescence is strongly enhanced (up to a factor of ten) as compared to single-layer samples. The authors ascribe this effect to stronger light absorption and waveguiding effects in the nano-fibers. The paper may be interesting to readers working in the field of electroluminescence of rare-earth-containing materials. It can be published, after the authors have addressed the following issues.
1. P. 2, line 67: "... electronic transition in the 4f band ..." (typo).
2. P. 2, line 79: The acronym "IMD software" should be explained.
3. P. 6, caption of Fig. 4: The acronym "FESEM" should be explained.
4. P. 10, line 281: The infrared region begins at 700 ... 800 nm, not at 200 nm.
5. P. 11, first paragraph: Some more details of the theoretical calculations should be given: Which parameters of the samples enter in the calculations (only the indices of refraction and the thickness or some additional parameters)? Was the doping with the Tm3+ ions taken into account? Where do the Tm3+ ions absorb (only in the UV or also in the visible range)? If in the visible range, why is their absorption not visible in the transmittance spectra?
6. Pp. 13, 14, Table 2: This table contains several flaws: a) Lorentzian linewidths on the order of 10-8, 10-40, or even 10-220 nm have no physical meaning. Write "0" instead. What does an error margin of 1.5952 x 10+7 nm mean when the average value is 2.9321 x 10-22 nm? b) Line positions or areas should be given with a reasonable number of digits. An area of, e.g., 241217,0418 units is useless, even when the computer yields this result. The last six or seven digits can be omitted. c) The unit cm2 for the area has no physical meaning. The area is proportional to the product of linewidth and peak intensity. If the peak intensity is given in arbitrary units (a.u.), then also the area can only be given in a.u.
7. The quality of the English is very poor and needs improvement.
Reviewer 3 Report
Dear authors,
I enjoyed reading your manuscript. I recommend it for publication. I have not so much an objection as a wish: was it possible to describe the theoretical calculations in more detail, and not just refer to the description of the software [20]? Also, I think that the plots in Figure 2a represent X-ray interference on thin films rather than diffraction (XRD pattern). I think that would be more correct.
Author Response
Response to Reviewer 3 Comments
I enjoyed reading your manuscript. I recommend it for publication. I have not so much an objection as a wish:
Point 1: Was it possible to describe the theoretical calculations in more detail, and not just refer to the description of the software [20]?
Response 1: Detail description of the theoretical calculation has been added as advised by a reviewer (page 4,line 142-155).
Point 2: Also, I think that the plots in Figure 2a represent X-ray interference on thin films rather than diffraction (XRD pattern). I think that would be more correct.
Response 2: Changes has been made (page 4, line 158).
Please see the attachment.
